# Targeting Autophagy in Breast Cancer

**DOI:** 10.3390/ijms21217836

**Published:** 2020-10-22

**Authors:** Stefania Cocco, Alessandra Leone, Michela Piezzo, Roberta Caputo, Vincenzo Di Lauro, Francesca Di Rella, Giuseppina Fusco, Monica Capozzi, Germira di Gioia, Alfredo Budillon, Michelino De Laurentiis

**Affiliations:** 1Breast Unit, Istituto Nazionale Tumori IRCCS “Fondazione G. Pascale”, Via Mariano Semmola 53, 80131 Napoli, Italy; m.piezzo@breastunit.org (M.P.); r.caputo@breastunit.org (R.C.); dilaurovincenzo87@gmail.com (V.D.L.); f.dirella@istitutotumori.na.it (F.D.R.); g.fusco@breastunit.org (G.F.); m.capozzi@istitutotumori.na.it (M.C.); germiradigioia@gmail.com (G.d.G.); 2Experimental Pharmacology Unit, Istituto Nazionale Tumori IRCCS “Fondazione G. Pascale”, Via Mariano Semmola 53, 80131 Napoli, Italy; a.leone@istitutotumori.na.it (A.L.); a.budillon@istitutotumori.na.it (A.B.)

**Keywords:** breast cancer, autophagy, ATG, Chloroquine, Hydroxychloroquine, ACD

## Abstract

Breast cancer is a heterogeneous disease consisting of different biological subtypes, with differences in terms of incidence, response to diverse treatments, risk of disease progression, and sites of metastases. In the last years, several molecular targets have emerged and new drugs, targeting PI3K/Akt/mTOR and cyclinD/CDK/pRb pathways and tumor microenvironment have been integrated into clinical practice. However, it is clear now that breast cancer is able to develop resistance to these drugs and the identification of the underlying molecular mechanisms is paramount to drive further drug development. Autophagy is a highly conserved homeostatic process that can be activated in response to antineoplastic agents as a cytoprotective mechanism. Inhibition of autophagy could enhance tumor cell death by diverse anti-cancer therapies, representing an attractive approach to control mechanisms of drug resistance. In this manuscript, we present a review of autophagy focusing on its interplay with targeted drugs used for breast cancer treatment.

## 1. Introduction

Autophagy literally means “self-eating”. It is a complex multi-faceted process in which a cell destroys old or defective cellular components, which can later be recycled by the cell to meet its metabolic needs [1]. In mammalian cells, autophagy can be classified into three main categories: macroautophagy, microautophagy, and chaperone-mediated autophagy. In particular, the lysosomal degradation pathway macroautophagy (henceforth, autophagy), represents an adaptive response to different forms of stress, such as nutrient deprivation, growth factor depletion, infection, hypoxia or cellular damages induced by chemical agents [2]. In this context, autophagy could exert cytoprotection, by selectively eliminate potential cytotoxic materials such as misfolding proteins or damaged mitochondria, in a process defined “selective autophagy”, or supplying nutrients and energy during fasting and other forms of stress [3]. In addition to the cytoprotective function, autophagy has been proposed as a mechanism of cell death, given that features of autophagy have been observed in dying cells: this has been referred to autophagic cell death (ACD) [4]. Accordingly, autophagic activity modulates many pathologies, including neurodegeneration, cancer, and infectious diseases [5,6]. During autophagy, cellular proteins and organelles undergo a catabolic process in which they are engulfed by autophagosomes, digested in lysosomes, and recycled to sustain cellular metabolism. In mammals, autophagy is regulated by ~16–20 high conserved autophagy-related genes (*ATG*), that control the process of autophagosome formation and its fusion with lysosomes. This process involves four steps, so-called initiation, nucleation, maturation, and degradation, in which different ATG proteins act at different levels [7]. The initiation and nucleation proteins promote the formation of the autophagic vesicle membrane, with the recruitment of ATGs proteins to a specific subcellular location termed the phagophore assembly site (PAS). Gradual elongation of the curved isolation membrane results in expansion of the phagophore into a double-membraned vesicle, termed the autophagosome, thereby trapping the engulfed cytosolic material as autophagic cargo. Then the autophagosome fuses with the lysosomal membrane to form an autolysosome, followed by the degradation of the autophagic body together with its cargo [8]. The main ATGs complexes, such as the Unc-51 like autophagy activating kinase (ULK1) complex, the Beclin–ATG4-1-VPS34 class III phosphoinositide 3-kinase (PI3K) complex, the ATG5-ATG12-ATG16 complex, and ATG8/LC3 complex, involved in this process, have been extensively reported in other review [9,10,11]. An overview of the autophagy process [12], its induction [13,14] and regulation [15] have been reported in Figure 1.

## 2. Main Mechanisms Inducing Autophagy

A major regulator of autophagy is the mammalian target of rapamycin (mTOR) pathway, which consists of 2 distinct signaling complexes known as mTORC1 and mTORC2, but only mTORC1 directly regulates autophagy. In high-nutrient conditions, ATG13 and ULK1 are both directly bound and phosphorylated by mTORC1 and remain inactive in this phosphorylated form. Upon amino-acids deprivation, mTOR repression shifts cellular metabolism toward autophagy and recycling of cytosolic constituents. mTORC1 sites on ULK1 are dephosphorylated and ULK1 dissociates from mTORC1. Concomitantly, ULK1 undergoes autophosphorylation, followed by phosphorylation of ATG13 and FIP200 [3,7,11].

A decrease in intracellular energy results in activation of homeostasis regulatory kinases 5′ AMP- activated protein kinase (AMPK), a serine/threonine-protein kinase acting as a sensor of low cellular energy levels as in glucose starvation. The activation of AMPK can induce the autophagic process through two different mechanisms: (1) inhibition of the mTOR protein kinase complex, or (2) by bypassing mTOR by directly phosphorylation of ULK1, VPS34 and Beclin1 [16,17].

In the last years, the research group of Ballabio et al. identified a new key regulator of autophagy in TFEB (transcription factor EB), which is a master transcription factor controlling the expression of target genes involved in autophagy. Also, TFEB is negatively regulated by mTORC1 and released upon starvation, when TFEB translocates from the cytoplasm (where it normally resides) to the nucleus where it is active and regulates the expression of its target genes involved in lysosomal biogenesis [14,18,19,20] (Figure 1).

For a long time, autophagy was considered a non-selective pathway activated in response to cellular stresses such as starvation or nutrient deprivation in order to provide essential amino acids and nutrients to sustain cellular growth. However, in the last years, extensive evidence has reported the key role of autophagy in selective degradation of dysfunctional organelles, protein aggregates, and intracellular pathogens [7,21]. Selective processes including removal of peroxisomes (pexophagy) [22] endoplasmic reticulum (ERphagy) [23], ribosomes (ribophagy) [24], lipid droplets (lipophagy) [25], invading microbes (xenophagy) [26], protein aggregates (aggrephagy) [27,28] and damaged mitochondria (mitophagy) [29] have been reported. As well, selective cargo receptors, such as p62/SQSTM 1, BNIP3L (BCL2-interacting protein 3-like) NBR1 (Next to BRCA1), CALCOCO2 (calcium-binding and coiled-coil domain-containing protein 2), and OPTN (optineurin), needed for selective degradation process, have been widely studied. Indeed, mutations in the molecules involved in selective autophagy have been associated with susceptibility to a variety of human diseases [7].

## 3. Role of Autophagy in Cancer Diseases

Autophagy has dual roles in cancer, acting both as tumor suppressor by preventing the accumulation of damaged proteins and organelles and as a mechanism of cell survival that can promote the growth of established tumors [30,31]. Selective autophagy, by inhibiting accumulation of oncogenic p62/SQSTM 1 protein aggregates and damaged organelles like mitochondria, exerts a key role in cellular quality control, reducing the production of reactive oxygen species (ROS), preventing chronic tissue damage and inflammation, and promoting genome stability [32,33]. In this regard, the knockdown of p62/SQSTM 1 in autophagy-defective cells prevented ROS and the DNAdamage response [34]. Autophagy defects, e.g., heterozygous knockdown Beclin1 and ATG7 in mice, are associated with susceptibility to metabolic stress, genomic damage, and tumorigenesis [35]. In contrast, excessive stimulation of autophagy due to Beclin1 overexpression can inhibit tumor development [36].

Other evidence indicates that the predominant role of autophagy in cancer cells is to confer stress tolerance, which serves to maintain tumor cell survival [37,38]. Cancer cells seem to activate autophagy to sustain high proliferation level that require high ATP demand. Cytotoxic and metabolic stresses, such as hypoxia and nutrient deprivation, can activate autophagy to recycle cellular components to sustain survival of cancer cells [37,39,40]. Elevated levels of autophagy have been observed in many tumors. In particular, Beclin1 was upregulated in colorectal cancer, gastric cancer, liver cancer, breast cancer, and cervical cancer [41,42,43,44], suggesting that the enhancement of autophagy can promote tumorigenesis. Inhibition of autophagy in tumor cells has been shown to increase the efficacy of anti-cancer drugs, supporting its key role in cancer [45,46]. Other findings reported that mutations in H-RAS or K-RAS in cancer cells can trigger a high level of autophagy [47,48], and in these cells, suppression of essential autophagy proteins inhibit cell growth, indicating that autophagy is an essential survival pathway in these tumors [49]. A relevant role of autophagy in cancer metastasis development has also been reported. This association was found in breast cancer metastasis [50,51], melanoma metastases [52], hepatocellular carcinoma [53], and glioblastoma [54]. In addition, autophagy seems to play a role in the survival of distant colonies in response to various environmental stress including hypoxia, nutrient deprivation, and detachment from the extracellular matrix (ECM) [55,56,57,58]. Beyond the role in tumorigenesis, supported by extensive evidence, autophagy may also play a role in cancer cell death and tumor suppression. The induction of ACD has been proposed as a mechanism of cell death, because accumulation of autophagosomes and autolysosomes have been observed in the cytoplasm of dying cells, without activation of apoptosis [59]. ACD could occur in conditions of prolonged stress, and consequently high activation of autophagy, which could cause an excessive protein and organelle turnover that overwhelm the capacity of the cell [60]. A form of ACD, called ‘‘autosis’’, has been reported during starvation in dying cells that showed not apoptosis or necrosis, but morphological characteristics of autophagy; this cell death could be prevented by inhibition of autophagy. [61,62]. In this context, an interplay between autophagy and apoptosis is supposed by the finding that in cancer models, when autophagy is inhibited, apoptosis is promoted [63]. Else, in preclinical models with defective apoptosis, other forms of cell death, such ACD, could occur [64,65,66,67,68,69,70,71,72].

### Targeting Autophagy as a Cancer Treatment

Because autophagy plays a role in tumor suppression, the modulation of autophagy may be an important strategy for cancer treatment. Several agents are being testing in preclinical and clinical trials, either to inhibit the cytoprotective role of autophagy or to induce ACD in apoptosis-resistant cells [73]. However, only some autophagy inhibitors have, so far, shown some efficacy in advanced cancer. Common autophagy- inhibiting molecules could be categorized into four groups according to their mode of action: (1) Repressors of autophagosome formation, class III PI3K inhibitors, e.g., 3-methyladenine (3-MA), Wortmannin, LY294002, SAR405 and Viridiol that block the formation of autophagosome. (2) Repressors of lysosomal acidification: Lysosomotropic agents, such as Cloroquine (CQ), Hydroxychloroquine (HCQ), Lys0569, and Monensin prevent acidification of lysosomes and thus inhibit degradation of the cargo in the autophagosomes; (3) Inhibitors of autophagosome-lysosome fusion: Vacuolar-ATPase inhibitors, Bafilomycin and Concanamycin variants that interfere with the fusion of autophagosomes with lysosomes; (4) Silencers of transcription of autophagy-related genes: it has been demonstrated, indeed, that by means of siRNA- or miRNA-mediated silencing strategies, knockdown of autophagy-related genes can be achieved with subsequent inhibition of autophagic activity [74,75,76,77,78,79].

In this regard, CQ and HCQ have been extensive studied as anti-cancer therapy, both in combination with conventional anti-cancer treatments or with new target-therapies, since they are able to sensitize tumor cells to a variety of drugs [80]. It has been fully described that CQ and HCQ exert anti-cancer effects due to their anti-autophagy activities. CQ and HCQ inhibit the autophagic flux at a late stage, increasing lysosomal pH, which inhibits lysosomal degradative enzymes. Moreover, other preclinical anti-cancer activities, like activation of TLR9/nuclear factor kappa B (NF-κB) signaling and p53 pathways, or inhibition of CXCL12/CXCR4 signaling pathway, have been described [80,81,82].

Adding CQ to chemotherapy increases sensitivity of various cancers, including lung cancer, ovarian cancer, glioma cancer, gastric cancer, bladder cancer, and endometrial cancer cells [83,84,85,86,87]. Else, autophagy seems to limit the therapeutic effects of PI3K/Akt/mTOR signaling inhibitors; therefore, autophagy inhibition can theoretically overcome tumor resistance to these agents: combination of mTORC1 inhibitors plus CQ, show a synergistic effect on cell death in different cancers such as renal cancer [88] colon cancer [89] hepatocellular carcinoma [90] and glioma [91]. In aggressive prostate cancers, the efficacy of the AKT inhibitor AZD5363 is limited, while the blocking of autophagy using 3-MA, CQ, and Bafilomycin enhance cell death [92]. Moreover, CQ has been shown to synergize with tyrosine kinase inhibitors such as Imatinib, Sorafenib, and Sunitinib by inducing apoptosis, in different preclinical models [93,94,95]. Beyond the speculative use of these drugs in preclinical studies, their use in clinical trials remains controversial. CQ is often cytotoxic at high doses and can promote cell cycle arrest [96] or DNA damage that induces cancerogenesis [97]. According to http://clinicaltrials.gov, there are several ongoing phase I/II trials evaluating the combination of HCQ/CQ alone or with chemotherapeutic and targeted agents in different cancers. The availability of clinical results is limited now, as most trials are still recruiting or ongoing, and those that have been completed focused primarily on safety and tolerability of CQ and HCQ in cancer.

## 4. Targeting Autophagy in Breast Cancer

Breast cancer is the most diagnosed cancer in women [98], characterized by different outcome depending on histopathological characteristics and molecular profile [99].

Based on gene expression profile, breast cancer can be classified into at least three subtypes: luminal tumors, which are positive for estrogen (ER+) and/or progesterone receptors (PR+); HER2-enriched, which overexpress the *ERBB2* oncogene; and triple-negative tumors (TNBC), which lack of hormone receptors and HER2 amplification. Each subtype has different characteristics, in terms of incidence, response to treatment, risk of disease progression, and sites of metastases [100]. Hormone receptor-positive (HR+) breast cancer subtypes (luminal A and B) represent approximately 60–75% of all breast cancers and respond well to endocrine therapy (ET), in both adjuvant and metastatic setting. However, the majority of these patients show de-novo resistance (primary) or develop acquired resistance (secondary) to ET, which requires the administration of sequential endocrine-based therapy, both as monotherapy and in combination with targeted therapy, before switching to chemotherapy-based regimens [101].

The HER2 subtype is associated with an aggressive behavior. However, this sub-group of patients could benefit from several anti-HER2 targeted therapies, including Trastuzumab, Pertuzumab, Trastuzumab emtansine (T-DM1), Lapatinib, Neratinib, etc. [102].

Finally, TNBC are more likely to exhibit an aggressive phenotype that become metastatic and resistant to various chemotherapeutics [103]. Atezolizumab, an immunocheckpoint inhibitor targeting the protein programmed cell death-ligand 1 (PDL1), was recently approved in combination with nab-paclitaxel, in the treatment of patients with unresectable locally advanced or metastatic PDL1-positive TNBC [104], however, chemotherapy is still the standard of care in many early or advanced TNBC tumors.

In the last years, preclinical and clinical research focused on targeting different pathways involved in tumor growth, such as PI3K/Akt/mTOR, cyclinD/CDK/pRb pathways and tumor microenvironment [103,105,106]. In this regard, inhibition of autophagy has proven to improve the drug response and reduce the mechanism of drug resistance [11,32,41].

### 4.1. ER+, PgR+, HER2-Subtype

ET with tamoxifen, a selective estrogen receptor modulator, or aromatase inhibitors (AIs) is recommended for HR+ breast cancer patients in pre and post-menopause [107]. Several mechanisms of resistance to ET have been identified [108]. In this regard, tamoxifen therapy has been reported to induce autophagy-mediated resistance in ER+ cellular model, MCF-7 and T-47D cells [109,110], while, in in vivo models, autophagy inhibitors restored antiestrogen sensitivity in resistant tumors [111].

At the same time, Exemestane (Exe), a powerful steroidal AI, promoted a cytoprotective autophagy in acquired resistant breast cancer cell models. In these models, the inhibition of autophagy and/or of PI3K pathway reverted Exe-resistance through apoptosis promotion, disruption of cell cycle, and inhibition of cell survival pathways [112]. Alterations of PI3K/AKT/mTOR pathway, such as gain-of-function mutations of the Phosphatidylinositol-4, 5-bisphosphate 3-kinase, catalytic subunit, alpha (PIK3CA) and loss/low expression of Phosphatase and Tensin Homolog (PTEN), could result in over-activation of this pathway, responsible of aberrant proliferation and cell survival [113]. In the last years, several PI3K/AKT/mTOR inhibitors were developed. It is well known that Everolimus, an mTOR inhibitor approved to be used in combination with Exemestane in a metastatic setting, could induce autophagy [112,114,115,116,117]. The increase in autophagy, induced by PI3K/AKT/mTOR inhibitors, is not surprising due to their inhibition of mTOR signaling, control of autophagy initiation and ULK1/2 phosphorylation. Recently, Alpelisib, an oral α-specific PI3K inhibitor, was approved, in metastatic setting [118], while other PI3K/AKT/mTOR inhibitors, such as Taselisib (PI3K inhibitor), Ipatasertib, and Capivasertib (AKT inhibitors) are in clinical investigation [103]. Recently, Zhai et al. reported that Ipatasertib was able to induce ACD in hepatocellular carcinoma models [119], while, Zorea et al. showed that an ovarian cancer cell line treated with Taselisib activates autophagy to avoid cell death [120].

Recently, CDK4/6 inhibitors in combination with ET have been approved both for AI-sensitive and AI-resistant patients [121]. Disregulation in cyclinD/CDK/Rb pathway is frequent in many types of human cancers, including breast cancer, particularly HR-positive. CDK4/6 kinases phosphorylate retinoblastoma protein (Rb) at serine 807/811, leading to E2F release, thus triggering G1--to--S transition. Currently, there are three approved oral highly selective CDK4/6 inhibitors, namely, Palbociclib (PD0332991), Ribociclib (LEE011), and Abemaciclib (LY2835219) [105,122]. These drugs have been widely studied in preclinical models. There is evidence that autophagy can occur after CDK4/6 inhibitors exposure in different cancer models with different response [30,123]: in hepatocellular carcinoma models, Palbociclib can induce apoptosis and ACD by activating AMPK [124]. In gastric cancer models, Palbociclib induced autophagy that occurred as an adaptive mechanism of cell survival in response to senescence, as the simultaneous blockade of CDK4/6 and autophagy exacerbated the senescence phenotype [125]. Else, in breast cancer models, cells activate autophagy in response to Palbociclib, and blockade of autophagy significantly improved the efficacy of CDK4/6 inhibition in in vitro and in vivo breast cancers models with an intact G1/S transition [126]. Other findings suggested that the CDK4 inhibitors could induce autophagy only in some cellular models of solid cancer, since autophagy induction was observed in selective cell lines, where the combination of CDK4 inhibitors and autophagy inhibitors induced apoptosis [127]. Also, Abemaciclib was reported to induce autophagy in multiple myeloma and renal cell carcinoma models [128,129]. However, it was recently reported that, in different cancer models, Abemaciclib induced an atypical cell death, independent from autophagy, accompanied by lysosomal dysfunction, block of autophagic flux, and accumulation of autophagosomes [130]. Recently, a relevant finding suggested that CDK4/6 kinases regulate lysosome biogenesis during cell cycle, through phosphorylation of TFEB/TFE3 in the nucleus, thereby promoting their shuttling to the cytoplasm. Otherwise, inhibition of CDK4/6 kinases increases lysosomal numbers by activating the lysosome and autophagy transcription factors TFEB and TFE3 [131].

### 4.2. HER2-Positive Subtype

HER2 is a transmembrane receptor tyrosine kinase of the epidermal growth factor receptor (EGFR) family. Its overexpression is associated with the hyperactivation of pathways, such as MAPK, JAK/STAT, RAS/MEK/ERK and PI3K/AKT/mTOR pathways, involved in cellular proliferation and differentiation of different cancers [132]. Anti-HER2 drugs could be divided into two classes: (1) HER2-directed antibodies, such as Trastuzumab, Pertuzumab and TDM1 and (2) small-molecule inhibitors targeting the kinase activity of the receptor, such as Lapatinib, Neratinib and Tucatinib [133]. Generally, HER2+ tumors have been shown to exhibit low levels of autophagy. A significant association between Beclin1 loss or decreased Beclin1 mRNA expression and HER2 amplification was found in breast cancer tumors [134,135]. In vitro studies confirmed that human and mouse HER2+ breast cancer cells presented low Beclin1 mRNA and a low autophagy-genes expression [136]. Finally, other evidence showed that the eHER2 receptor directly represses autophagy by interaction with Beclin1 [137]. At the same time, the signaling cascade activated by HER2 also activates mTORC1 signaling, a negative regulator of autophagy. Therefore HER2-target therapy has been reported to induce autophagy and its inhibition could be used to reduce the drug resistance [138]. In different preclinical models, mechanisms of resistance to Trastuzumab were associated to increased autophagy pathway [139]; while in Lapatinib resistant models, the inhibition of autophagy led to reversion of resistance [140,141]. Furthermore, Trastuzumab-resistant tumors present a higher level of AMPK expression [142].

### 4.3. TNBC Subtype

As mentioned above, TNBC tumors are characterized by a more aggressive behavior and early relapse [103]. Higher levels of basal autophagy were found in the metastatic cell lines when compared to the non-metastatic, suggesting that autophagy could promote invasiveness and possibly increase tolerance to the cellular stress occurring during the metastatic process [143]. TNBC tumors are more hypoxic than non-TNBC and it has been suggested that they are less sensitive to hypoxic conditions because of perpetually higher levels of autophagy [144]. Indeed, several reports have indicated that TNBC tumors exhibit a higher level of autophagy than other breast cancer subtypes. Expression of the autophagy-related microtubule-associated proteins, including Beclin1, LC3A and LC3B are higher in TNBC cells compared to the other breast cancer subtypes, with the lowest expression in the stroma of TNBC [145]. High expression of LC3B has also been associated with tumor progression and poor outcome in TNBC [51], demonstrating its role as a potential prognostic marker in TNBC. High expression of Autophagy marker ATG9 was found in TNBC breast cancer tissue, while its inhibition led to the inhibition of pro-cancer phenotypes [146]. Furthermore, knockdown of autophagy-related genes (LC3 and Beclin1) inhibited autophagy and significantly suppressed cell proliferation, colony formation, migration/invasion and induced apoptosis in MDA-MB-231 and BT-549 TNBC cells [147], while silencing of ATG5, ATG7 and Beclin1 reduced the proliferation of different TNBC cell lines (basal and claudin-low) [148]. Else, In 4T1 and 67NR mouse mammary tumor cells, CQ increased sensitivity to the mTOR inhibitor rapamycin and to the PI3K inhibitor LY294002, even in the absence of Beclin1 and ATG12, suggesting that CQ could also exert tumor growth suppression beyond autophagy [149]. Recently, it was reported an association between the hippo signaling pathway yes-associated protein (YAP) and autophagy in TNBC. In particular, autophagy induction promoted the YAP nuclear translocation and activation in TNBC models [150]. Overall, these data strongly suggest that autophagy is essential to the survival of TNBC cells and that blocking autophagy would lead to cell death [144,151].

Recently, the immunocheckpoint inhibitor Atezolizumab, a monoclonal antibody against PDL1, was approved for treatment in metastatic PDL1-positive TNBC [104]. Interestingly, emerging evidence points to the prominent role of autophagy in tumor immunoescape by multiple overlapping mechanisms [152]. Some evidence has shown an inter-play between autophagy and PDL1 expression: a study by Clark et al. showed that tumor-intrinsic PDL1 signals regulate cell proliferation and autophagy in ovarian cancer and melanoma. Tumor cells with high levels of PDL1 expression were more sensitive to autophagy inhibitors than cells with lower PDL1 [153]; in gastric cancer, autophagy regulates PDL1 expression through the p62/SQSTM1-NF-κB pathway, while pharmacological inhibition of autophagy increased the levels of PDL1 in gastric cancer cells and in xenografts models, influencing the therapeutic efficacy of PDL1 inhibitors [154]. These data suggest that autophagy could represent an attractive future therapeutic target to develop innovative and effective cancer immunotherapeutic approaches.

Extensive evidence highlighted the contribution of cancer stem cells (CSCs) in tumorigenic potential, high risk of metastasis, and drug resistance of TNBC [155]. CSCs represent a small population of cancer cells with staminal phenotype; they expose CD44+/CD24- and high ALDH expression [156,157,158]. This signature is associated with high capacity of self-renewal, of proliferation, and mammalian spheroids forming [159,160,161]. These CD44+/CD24- cells show an EMT phenotype, with a great tumorigenic potential and invasion and metastasis ability [156,162,163], that is responsible of capacity of CSCs to survive in hard metabolic conditions [164,165,166]. A relevant feature of CSCs is their ability to obtain energy from different sources and, therefore, to survive in hostile microenvironments, or in unfavorable circumstances encountered during tumor progression and in metastatic sites. Hypoxia-inducible factor-1 (HIF-1) is responsible of response to hypoxia in cancer cells [167,168], and it has been reported that the expression of EMT and stemness activators such as WNT, Hedgehog, and NOTCH pathways, or stemness markers such as FOXA2, cMET, CD133, NANOG, SOX2, SOX17, and PDX1 could also occur through the activation of HIF-1 [169,170]. Indeed, the exposure of TNBC cells to hypoxia increased the percentage of CSCs in a HIF-1–dependent manner [171,172]. Another study reported that in paclitaxel- or gemcitabine-treated TNBCs, the induction of HIF was required for enrichment of CSCs both in vitro and in vivo [169], while, in patients with breast cancer, HIF-1 overexpression was associated with increased mortality [173], poor prognosis [174], and drugs resistance [175]. It is well known that hypoxia induces autophagy, mediated by HIF-1α [176]. In this regard, autophagy has been shown to impact on CSCs generation, differentiation, plasticity, migration/invasion, and drug resistance [177]. Autophagy has been related to a variety of CSCs in several cancers, including breast, and its inhibition affected the expression of staminal markers and the cell self-renewal capacity [178,179]. The expression of Beclin1 and ATG4 was found upregulated in mammospheres and needed for their maintenance and expansion [179]. In two breast cancer stem-like cell lines, autophagy was found act through EGFR/Stat3 and Tgfβ/Smad signaling, to induce stem phenotype [180]. In CSCs originated by TNBC, autophagy inhibition decreased the secretion of IL-6, through the STAT3/JAK2 pathway, impairing their maintenance and stem phenotype [148]. While, inhibition of autophagy by CQ impaired cell migration and invasion, increased expression of the epithelial marker CD24, and decreased vimentin in CSCs [181].

### 4.4. Clinical Trials

Several clinical trials are investigating the use of CQ or HCQ in different cancer types. Results from early phase clinical trials with CQ or HCQ have been, so far, controversial [182,183,184,185]. A recent meta-analysis of clinical trials evaluating treatment with HCQ alone or in combination with chemotherapy or radiation in different cancers, showed that the addition of autophagy-inhibitor-based therapy has a better treatment response compared to chemotherapy or radiation alone. In particular, autophagy inhibition exerted the best survival benefit in patients with glioblastoma [186].

On website clinicaltrials.gov, about 11 clinical trials are reported to investigate the use of CQ or HCQ with different dosage, alone or in addition to other therapies, in breast cancer (Table 1). Recently, Arnaout et al. (NCT02333890) published the results from a randomized, double-blind clinical trial evaluating the effects of treatment with single-agent CQ 500 mg daily for 2- to 6-weeks prior to breast surgery. Results showed that in the preoperative setting, the treatment was not associated with significant effects on breast cancer cell proliferation, while it was associated with toxicity that may affect its broader use in oncology [187]. The disappointing results achieved in clinical trials could be in part ascribed to the choose of the wrong target. Else, the clinical usefulness of CQ may have been limited because high doses are required to compensate for its nonselective distribution in vivo, with high intersubject variability in the steady state blood concentrations of CQ and accumulation of its metabolites, desethylchloroquine, and bisdesethylchloroquine, responsible for related adverse effects [188,189].

## 5. Conclusions

Autophagy plays a complex role in cancer, and its inhibition could, in theory, represent an effective therapeutic strategy. In the last years, new targeted drugs, such as CDK4/6 inhibitors, PI3K/AKT inhibitors, new anti-HER2 drugs and immunocheckpoint inhibitors have emerged and have been integrated in the clinical practice. We have reviewed how these therapies could modulate autophagy. We have also highlighted how this modulation may play a key role in the mechanisms of resistance to these drugs. In this perspective, inhibition of autophagy may help reversing or overcoming drug resistance. However, clinical applications with autophagy inhibitors, have been so far disappointing, due either to the non-efficient in vivo autophagy inhibition or to the toxicity of the tested drugs.

Based on preclinical data, however, targeting autophagy remains a promising approach in breast cancer, and the development of new agents with a higher therapeutic index should be regarded as a worthwhile achievement in pharmacological research.

## Figures and Tables

**Figure 1 ijms-21-07836-f001:**
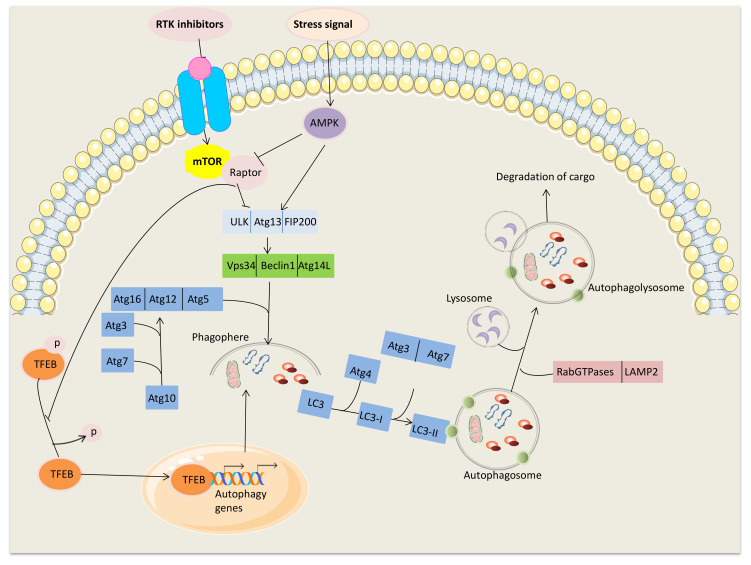
The Unc-51 like autophagy activating kinase (ULK1) serine threonine kinase complex, involving ULK1, FIP200, and ATG13, acts in the initiation process, phosphorylating multiple downstream factors. Then, the complex involving Beclin1, VPS34, a class III phosphatidylinositol-3-kinase (PI3K) and ATG14 acts in autophagosome nucleation step. During the maturation step, the ATG7 and ATG10 conjugate ATG5 to ATG12, and ATG7 and ATG3 conjugate microtubule-associated protein 1 light chain 3 (LC3) (ATG8) to the lipid phosphatidylethanolamine (PE). The ATG5-ATG12 conjugate forms a complex with ATG16, and the ATG5-ATG12-ATG16 complex gets anchored onto phosphoinositol 3-phosphate generated by VPS34 on emerging autophagosomal membranes through WD repeat domain, phosphoinositide-interacting 2b (WIPI-2b). The complex is efficient for the LC3 conjugation system. LC3 is first conjugated with PE. ATG4 plays a role in lipoxidating LC3 to LC3-I and exposes the C-terminal glycine of LC3 for the subsequent conjugation of PE. PE is conjugated to the C-terminal glycine of LC3-I, and this conjugation needs to be catalyzed by the E1-like enzyme ATG7 and the E2-like enzyme ATG3. The autophagosomal membrane is conjugated with LC3-PE. During fusion with the lysosome, the LC3-II bound to the outer membrane is cleaved and recycled by ATG4, while LC3-PE associated with the inner membrane is degraded by lysosomal proteases along with the cargo of the autophagosome, thus recycling amino acids, fatty acids, and nucleotides.

**Table 1 ijms-21-07836-t001:** Clinical trials in breast cancer with Chloroquine or Hydroxychloroquine by clinicaltrial.gov website.

Breast Cancer Stage/Subtype/Setting	Intervention	Study Design	Dosage	Clinicaltrial.gov
Invasive Breast Cancer/Neoadjuvant	Drug: Chloroquine Drug: Placebo	Phase 2 Randomized, Double-blind, placebo-controlled	chloroquine 500 mg daily as an oral capsule during the wait time to surgery.	* NCT02333890
Advanced or Metastatic Breast Cancer	Chloroquine in Combination with taxane or taxane-like Chemo Agents	Phase II Non-Randomized, open label	chloroquine 250 mg daily as an oral capsule	NCT01446016
Carcinoma, Intraductal, Noninfiltrating DCIS Ductal Carcinoma In Situ Neoadjuvant	Chloroquine	Phase 1 Phase 2 Non-Randomized open label	chloroquine standard dose (500mg/week) or chloroquine low dose (250mg/week) for 1 month prior to surgical removal of the tumor.	NCT01023477
Metastatic Breast Cancer	Hydroxychloroquine + Ixabepilone	Phase 1 Phase 2 Non-Randomized, open label	Dose escalation from 200 mg po qd to 200 mg po bid.	# NCT00765765
Estrogen Receptor-Positive Metastatic Breast Cancer	Hydroxychloroquine in combination with the current hormonal therapy	Phase Ib/II Study Non-Randomized, open label	Not available	NCT02414776
Invasive breast cancer	Arm A: Abemaciclib Arm B: Abemaciclib + Hydroxychloroquine	Phase II Pilot Trial, Randomized, open lable	Hydroxychloroquine 600 mg BID	NCT04523857
Recurrent Breast Cancer	Phase II Arm A: Hydroxychloroquine for 24 weeks Arm B: Hydroxychloroquine for 24 weeks + Gedatolisib and × 2 weeks administered weekly Arm C: Hydroxychloroquine for 24 weeks + Gedatolisib and × 6 weeks administered weekly Arm D: Hydroxychloroquine for 24 weeks + Gedatolisib and × 12 weeks administered weekly Phase Ib: Hydroxychloroquine for 24 weeks + Gedatolisib and × 6 weeks administered weekly	Phase 1 Phase 2 Randomized open label	Hydroxychloroquine 600 mg BID	NCT03400254
Breast Cancer Stage IIB	Arm A: Hydroxychloroquine Arm B: Everolimus Arm C: Hydroxychloroquine + Everolimus Arm D: Observation	Phase 2 Pilot Randomized	Not available	NCT03032406
HR+/Her 2- Advanced Breast Cancer	Arm A: Abemaciclib + Hydroxychloroquine 200 mg b.i.d. Arm B. Abemaciclib + Hydroxychloroquine 400 mg b.i.d. Arm C: Abemaciclib + Hydroxychloroquine 600 mg b.i.d. Arm D: Abemaciclib + Hydroxychloroquine + endocrine therapy.	Phase 1 Non-Randomized Sequential Assignment Open Label	Hydroxychloroquine 200 mg Hydroxychloroquine 400 mg Hydroxychloroquine 600 mg	NCT04316169
Participants with advanced, metastatic (stage IV) breast cancer (phase I) Participants with early stage (stage I-III) breast cancer (phase II)	Hydroxychloroquine, Palbociclib, and Letrozole	Phase 1 Phase 2 Open label	Hydroxychloroquine 400 mg	NCT03774472
Breast Cancer Neoadijuvant	Hydrochloroquine	Phase II Pilot Open Label	Hydrochloroquine 800 mg per os once, and then 400 mg per day	NCT01292408

* This study achieved negative results [187]. # This study achieved a Response Rate of 30%.

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
