# Peer review of "Targeting Autophagy in Breast Cancer"

_ijms, 2020, doi:10.3390/ijms21217836_

Round 1
Reviewer 1 Report
The review is very interesting and very good structurate. I suggest few minor language and editing corrections.
Line 35. In this sentence the author's should change categorize with classificated. In mammalian cells, autophagy can be categorized into three main ways:
Line 36. The sentence is too long and need to be rearrange. Especially Of these, shoud be replaced.
Line 49. ~16-20 high conserved autophagy-related genes (ATG)-one space less
Line 51. Specified this abbreviation because it's first time that you mention it- the ULK1 kinase complex
Line 80 These subtitle needs to be rearrange- 2. Main mechanisms of induction of Autophagy
Line 122. Initiation of the sentence needs to be rearrange with some synonyms because it's not clear. Through its role in quality control,
Line 148 3.1 Autophagic cell death and line 3.2 Targeting autophagy for cancer prevention-these paragraphs should be fusiused because they are too dispersive and long. The main goal of massage should be more concrete.
Line 245. The sentence is too confused so you need to rearrange it.
Line 398. Can you maybe improve the conclusion message here and include a statement or your interpretation. These two sentence are very hard to follow and do not sand a right and critical overview to the scientific community.
New targeted drugs for breast cancer, such as CDK4/6 inhibitors,
PI3K/AKT inhibitors, new anti-HER2 drugs and immunocheckpoint inhibitors, may induce, or interfere with, autophagy and we have reviewed how autophagy may play a key role in the mechanisms of resistance to these drugs. Clinical applications of autophagy inhibition, however, have been so far disappointing, due either to the non-efficient in vivo autophagy inhibition or to the toxicity of the tested drugs.
Author Response
The review is very interesting and very good structurate. I suggest few minor language and editing corrections.
Line 35. In this sentence the author's should change categorize with classificated. In mammalian cells, autophagy can be categorized into three main ways:
Updated
Line 36. The sentence is too long and need to be rearrange. Especially Of these, shoud be replaced.
Updated
Line 49. ~16-20 high conserved autophagy-related genes (ATG)-one space less
Updated
Line 51. Specified this abbreviation because it's first time that you mention it- the ULK1 kinase complex
Updated
Line 80 These subtitle needs to be rearrange- 2. Main mechanisms of induction of Autophagy
We changed the title in Main mechanisms inducing Autophagy
Line 122. Initiation of the sentence needs to be rearrange with some synonyms because it's not clear. Through its role in quality control,
We changed the sentence, please refer to line 114,115
Line 148 3.1 Autophagic cell death and line 3.2 Targeting autophagy for cancer prevention-these paragraphs should be fusiused because they are too dispersive and long. The main goal of massage should be more concrete.
We reduced the paragraph 3.1 and fused it with 3., while we suppose that paragraph 3.2 should be separated because it addressed a different and relevant topic.
Line 245. The sentence is too confused so you need to rearrange it.
Updated
Line 398. Can you maybe improve the conclusion message here and include a statement or your interpretation. These two sentence are very hard to follow and do not sand a right and critical overview to the scientific community.
New targeted drugs for breast cancer, such as CDK4/6 inhibitors,
PI3K/AKT inhibitors, new anti-HER2 drugs and immunocheckpoint inhibitors, may induce, or interfere with, autophagy and we have reviewed how autophagy may play a key role in the mechanisms of resistance to these drugs. Clinical applications of autophagy inhibition, however, have been so far disappointing, due either to the non-efficient in vivo autophagy inhibition or to the toxicity of the tested drugs.
We changed some sentences in conclusions, as you suggested, to clarify the final message of this review.
Reviewer 2 Report
Targeting autophagy in breast cancer
Authors have wisely discussed the autophagic role in cancer; particularly in breast cancer. Their description provides a clear picture of both autophagy induction and inhibition capacities in breast cancer. Despite the therapeutic benefit of chemotherapeutic agents, their resistance caused by autophagy dysregulation are well captured.
The manuscript would be benefitted more if authors consider the minor corrections, which mentioned below.
Line 46: should be “including neurodegeneration, cancer, infectious diseases, autoimmune disease, and others (https://doi.org/10.1016/j.jaut.2018.08.009)”.
Figure 1 shows mTOR, TFEB, and LAMP2 pathways. However, the description in the figure legend in missing. Or else, appropriate references should be cited for further reading.
Line 70: phosphatidylethanolamine → phosphatidylethanolamine (PE)
Line 73: microtubule-associated protein 1 light chain 3 (LC3) → LC3
Line 74: phosphatidylethanolamine (PE) → PE
Line 77: the outer membrane is cleaved → the LC3-II bound to the outer membrane is cleaved
Line 72-79: For more information….please refer and cite this review from MDPI, https://doi.org/10.3390/cells9051321
Line 93: Beclin 1 or Beclin-1 or Beclin1…… please be uniform throughout the manuscript.
Line 98: its target genes ….what are they…. Please refer this and cite https://doi.org/10.1126/science.1174447
Line 109: erphagy → ERphagy
Line 110: protein aggregates → protein aggregates (aggrephagy)
Line 159: other forms of cell death such ACD, → other forms of cell death, such as ACD,
Line 164, 166: cancer prevention → cancer treatment
Line 170: class III PI3K inhibitors 3-methyladenine → class III PI3K inhibitors, e.g., 3-methyladenine
Line 172: Lysosomotropic agents such as → Lysosomotropic agents, such as
Punctuation errors are prevalent throughout the manuscript. Please correct them.
Anti-cancer (line 180, 181, 185, and other places) or anticancer (line 183, and other placess)… be uniform throughout.
Line 184: PH → pH
Line 184: typo “enzimes”
Line 185: other preclinically → other preclinical
Line 188: Adding CQ to chemotherapy increases sensitivity of various cancers to cytotoxic drugs, including lung cancer, → Adding CQ to chemotherapeutic drugs increases sensitivity of various cancers, including lung cancer,
Line 197: in different in preclinical models → in different preclinical models
Line 213: triple negative tumors (TNBC) → triple-negative breast cancer (TNBC)
Line 236: Endocrine therapy (ET) → ET
PI3K/AKT/mTOR or PI3K-AKT/mTOR; cyclin/CDK/Rb or cyclin D/CDK/pRb …. Please be uniform
Line 286: HER2 (also known as ERBB2)… this should be explained earlier in page 6 (section 4), where it appears first.
Line 300: HER2 also activates mTOR1 or mTORC1?
Line 313: autophagy that other BC subtypes → autophagy than other breast cancer subtypes
Once abbreviated, please use acronym throughout the manuscript…e.g., chloroquine
Line 358: in vitro and in vivo → in vitro and in vivo
Line 389, 400: “in vivo” should be in italics
Table 1: results column is redundant; as only two treatment categories declared the results. Use the table legend to explain the same.
Author Response
Authors have wisely discussed the autophagic role in cancer; particularly in breast cancer. Their description provides a clear picture of both autophagy induction and inhibition capacities in breast cancer. Despite the therapeutic benefit of chemotherapeutic agents, their resistance caused by autophagy dysregulation are well captured.
The manuscript would be benefitted more if authors consider the minor corrections, which mentioned below.
Line 46: should be “including neurodegeneration, cancer, infectious diseases, autoimmune disease, and others (https://doi.org/10.1016/j.jaut.2018.08.009)”.
Updated, we inserted the indicated reference (6)
Figure 1 shows mTOR, TFEB, and LAMP2 pathways. However, the description in the figure legend in missing. Or else, appropriate references should be cited for further reading.
We updated the references 12- 15
Line 70: phosphatidylethanolamine → phosphatidylethanolamine (PE) corrected
Line 73: microtubule-associated protein 1 light chain 3 (LC3) → LC3 corrected
Line 74: phosphatidylethanolamine (PE) → PE corrected
Line 77: the outer membrane is cleaved → the LC3-II bound to the outer membrane is cleaved
corrected
Line 72-79: For more information….please refer and cite this review from MDPI, https://doi.org/10.3390/cells9051321
Updated, we inserted the indicated reference (12)
Line 93: Beclin 1 or Beclin-1 or Beclin1…… please be uniform throughout the manuscript.
corrected
Line 98: its target genes ….what are they…. Please refer this and cite https://doi.org/10.1126/science.1174447
Updated, we inserted the indicated reference (20)
Line 109: erphagy → ERphagy Updated
Line 110: protein aggregates → protein aggregates (aggrephagy) Updated
Line 159: other forms of cell death such ACD, → other forms of cell death, such as ACD, corrected
Line 164, 166: cancer prevention → cancer treatment corrected
Line 170: class III PI3K inhibitors 3-methyladenine → class III PI3K inhibitors, e.g., 3-methyladenine corrected
Line 172: Lysosomotropic agents such as → Lysosomotropic agents, such as corrected
Punctuation errors are prevalent throughout the manuscript. Please correct them. Updated
Anti-cancer (line 180, 181, 185, and other places) or anticancer (line 183, and other placess)… be uniform throughout. Updated as Anti-cancer
Line 184: PH → pH Updated
Line 184: typo “enzimes” Updated
Line 185: other preclinically → other preclinical corrected
Line 188: Adding CQ to chemotherapy increases sensitivity of various cancers to cytotoxic drugs, including lung cancer, → Adding CQ to chemotherapeutic drugs increases sensitivity of various cancers, including lung cancer,
corrected
Line 197: in different in preclinical models → in different preclinical models corrected
Line 213: triple negative tumors (TNBC) → triple-negative breast cancer (TNBC) updated
Line 236: Endocrine therapy (ET) → ET
corrected
PI3K/AKT/mTOR or PI3K-AKT/mTOR; cyclin/CDK/Rb or cyclin D/CDK/pRb …. Please be uniform
Updated PI3K/AKT/mTOR and cyclin D/CDK/pRb
Line 286: HER2 (also known as ERBB2)… this should be explained earlier in page 6 (section 4), where it appears first.
we indicated “HER2-enriched, which overexpress the ERBB2 oncogene” line 203
Line 300: HER2 also activates mTOR1 or mTORC1? Corrected to mTORC1
Line 313: autophagy that other BC subtypes → autophagy than other breast cancer subtypes
corrected
Once abbreviated, please use acronym throughout the manuscript…e.g., chloroquine
updated
Line 358: in vitro and in vivo → in vitro and in vivo
updated
Line 389, 400: “in vivo” should be in italics
updated
Table 1: results column is redundant; as only two treatment categories declared the results. Use the table legend to explain the same.
Updated as you suggested